# Levitation Characteristics Analysis of a Diamagnetically Stabilized Levitation Structure

**DOI:** 10.3390/mi12080982

**Published:** 2021-08-19

**Authors:** Shuhan Cheng, Xia Li, Yongkun Wang, Yufeng Su

**Affiliations:** School of Mechanical and Power Engineering, Zhengzhou University, Zhengzhou 450001, China; ritchie1028@163.com (S.C.); Lx2007@zzu.edu.cn (X.L.); Wangyongkun1999@163.com (Y.W.)

**Keywords:** diamagnetically stabilized levitation, Taguchi method, stable levitation, maximum gap

## Abstract

A diamagnetically stabilized levitation structure is composed of a floating magnet, diamagnetic material, and a lifting magnet. The floating magnet is freely levitated between two diamagnetic plates without any external energy input. In this paper, the levitation characteristics of a floating magnet were firstly studied through simulation. Three different levitation states were found by adjusting the gap between the two diamagnetic plates, namely symmetric monostable levitation, bistable levitation, and asymmetric monostable levitation. Then, according to experimental comparison, it was found that the stability of the symmetric monostable levitation system is better than that of the other two. Lastly, the maximum moving space that allows the symmetric monostable levitation state is investigated by Taguchi method. The key factors affecting the maximum gap were determined as the structure parameters of the floating magnet and the thickness of highly oriented pyrolytic graphite (HOPG) sheets. According to the optimal parameters, work performance was obtained by an experiment with an energy harvester based on the diamagnetic levitation structure. The effective value of voltage is 250.69 mV and the power is 86.8 μW. An LED light is successfully lit on when the output voltage is boosted with a Cockcroft–Walton cascade voltage doubler circuit. This work offers an effective method to choose appropriate parameters for a diamagnetically stabilized levitation structure.

## 1. Introduction

Diamagnetism is a natural property of a substance and exists in all materials. However, it is not easily appreciable in daily life, because it is too weak, compared to magnetism and paramagnetism. To observe diamagnetism, the diamagnetic material needs to be placed in a strong external magnetic field [1]. In an external magnetic field, the diamagnetic material generates a weak magnetic field, which is opposite to the external magnetic field. As a result, the diamagnetic material is subjected to a repelling force from the external magnetic field. When the repelling force and gravity of the diamagnetic material are equal and opposite to each other, the diamagnetic material is levitated in the external magnetic field, which is known as diamagnetic levitation. In 1939, diamagnetic levitation [2] was first observed by experiments where a small piece of bismuth and graphite was freely levitated in a strong electromagnetic field. In 2000, Simon et al. [3] further studied diamagnetic levitation and proposed diamagnetically stabilized levitation, which is a variant of diamagnetic levitation. In the study, the magnet served as a floater and was stably levitated between the diamagnetic materials without any external energy input.

In recent years, some applications based on diamagnetically stabilized levitation have been reported, such as sensors [4,5,6], actuators [7], and vibration energy harvesters [8,9,10]. Hilber et al. [11] presented a sensor based on diamagnetically stabilized levitation, which can be used to measure the density and viscosity of fluids in microfluidic systems. Ye et al. [12] designed a vibration energy harvester using diamagnetically stabilized levitation to harvest ambient vibration energy. Ding et al. [13] from the same research group conducted simulation and analysis on the energy harvester, and conducted experimental tests on a set of structural parameters. Clara et al. [14] investigated a viscosity and density sensor using diamagnetically stabilized levitation of a floater magnet on pyrolytic graphite. Liu et al. [15] studied a diamagnetically levitated electrostatic micromotor, which were fabricated by MSMS process and precision machining. Gisela et al. [16] constructed a low-cost magnetic levitation system. Chow et al. [17] studied the shape effect of magnetic sources formed by standard coil and ring magnet elements on diamagnetically stabilized levitation.

In this paper, by studying the static levitation characteristics of a structure constructed by Ding et al. [13], the levitation characteristics of the floating magnet in diamagnetically stabilized levitation are analyzed by simulation and experiments, and it was found that the floating magnet has three different levitation states, namely symmetric monostable levitation, bistable levitation, and asymmetric monostable levitation. Three levitation states were obtained by adjusting the gap of the diamagnetic materials. In order to make the energy harvester have better output characteristics, the moving space of the floating magnet is introduced, and the increase of this parameter is conducive to the arrangement of more coils. The maximum moving space that allows the floating magnet to achieve symmetric monostable levitation is determined by the structure parameters of the diamagnetically stabilized levitation. The influence of the structure parameters on the maximum moving space was studied by the Taguchi method. It was found through experiments that this method can effectively optimize the selection of structural parameters and improve the output characteristics.

## 2. Theory of Diamagnetically Stabilized Levitation

The structure of the diamagnetically stabilized levitation is shown in Figure 1a, which consists of a lifting magnet, an upper HOPG sheet, a floating magnet, and a lower HOPG sheet. The floating magnet is stably levitated between the two HOPG sheets.

The potential energy of the floating magnet in the field of the lifting magnet can be written as follows [3]:(1)$$U=-\vec{M}\cdot \vec{B}+mgz=-MB+mgz$$
where $\vec{M}$ and *m* are the magnetic dipole moment and mass of the floating magnet, respectively, *g* is the gravity acceleration, *B* is the magnetic flux density of the lifting magnet, *z* is the distance of the magnet orthogonal to the reference surface. With magnetic torques, the floating magnet aligns with the local field direction. As a result, energy only relies on the magnitude of the magnetic field.

Expanding the field magnitude of the lifting magnet around the levitation position in polar coordinates and adding two new terms $C_{z}z^{2}$ and $C_{r}r^{2}$ which denote the effect of diamagnetic materials, the potential energy of the floating magnet can be rewritten as:(2)$$U=-M\left[B_{0}+\left\{B^{\prime }-\frac{mg}{M}\right\}z+\frac{1}{2}B^{\prime \prime }z^{2}+\frac{1}{4}\left\{\frac{{B^{\prime }}^{2}}{2B_{0}}-B^{\prime \prime }\right\}r^{2}+\cdots \right]+C_{z}z^{2}+C_{r}r^{2}$$
where $B^{\prime }=\frac{\partial B_{z}}{\partial z}$ and $B^{\prime \prime }=\frac{\partial^{2}B_{z}}{\partial z^{2}}$.

The expression in the first curly bracket must be equal to zero when the floating magnet locates at the levitation position. In other words, the gravity of the floating magnet is balanced by the force derived from the non-uniform magnetic field:(3)$$B^{\prime }=\frac{mg}{M}$$

Furthermore, the conditions for vertical stability and horizontal stability can be derived according to Equation (2):(4)$$K_{v}\equiv C_{z}-\frac{1}{2}{\mathrm{MB}}^{\prime \prime }>0\;\;\mathrm{Vertical}\;\mathrm{stability}$$
(5)$$K_{h}\equiv C_{r}+\frac{1}{4}M\left\{B^{\prime \prime }-\frac{{B^{\prime }}^{2}}{2B_{0}}\right\}=C_{r}+\frac{1}{4}M\left\{B^{\prime \prime }-\frac{m^{2}g^{2}}{2M^{2}B_{0}}\right\}>0\;\mathrm{Horizontal}\;\mathrm{stability}$$

To achieve a stable levitation for the floating magnet, these conditions are necessary to ensure a local minimum of *U* at the equilibrium point. When Equations (4) and (5) are fulfilled, the stable levitation is possible if $MB^{\prime }=mg$. Therefore, the condition can be matched by adjusting the field gradient or the weight of the floating magnet.

In addition, the energy generated by two HOPG sheets can be expressed as [18]:(6)$$U_{dia}=C_{z}z^{2}=\frac{6\mu_{0}M^{2}\left|\chi \right|}{\pi L_{2}{}^{5}}z^{2}$$
where *L*_2_ is the gap between two HOPG sheets, *χ* is the magnetic susceptibility of the diamagnetic material.

According to Equations (4)–(6), the condition of the stable levitation can be obtained at the point where $B^{\prime }=mg/M$, which can be written as follows:(7)$$\frac{12\mu_{0}M\left|\chi \right|}{\pi L_{2}{}^{5}}>B^{\prime \prime }>\frac{{\left(mg\right)}^{2}}{2M^{2}B_{0}}$$

This puts a limit on the gap *L*_2_ [17]:(8)$$L_{2}<{\left\{\frac{12\mu_{0}M\left|\chi \right|}{\pi B^{\prime \prime }}\right\}}^{1/5}<{\left\{\frac{24\mu_{0}B_{0}M^{3}\left|\chi \right|}{\pi {\left(mg\right)}^{2}}\right\}}^{\frac{1}{5}}$$

It can be seen that the gap *L*_2_ should be limited in a certain range for stabilizing the floating magnet. However, the levitation characteristic of a floating magnet has not been discussed with different gap *L*_2_.

## 3. Analysis of Levitation Characteristics

To understand the levitation characteristic, the mechanics analysis of the floating magnet is performed, which is shown in Figure 1b. Since the magnetization directions of the two magnets are the same, an upward magnetic traction *F_m_* is exerted on the floating magnet by the lifting magnet. In addition, two opposite repulsive forces (*F_u_* and *F_l_*) generated by two HOPG sheets simultaneously act on the floating magnet. Therefore, the resultant force *F_r_* exerted on the floating magnet can be written as follows:(9)$$F_{r}=F_{m}+F_{l}-F_{u}-G$$
where *G* is the gravity of the floating magnet.

When the floating magnet is levitated at an equilibrium position, the resultant force is equal to zero. As shown in Figure 2, finite element analysis (FEA) simulation was performed by COMSOL Multiphysics 5.5, so as to obtain the resultant force. The structure parameters used in the simulation are listed in Table 1, and the simulation results are shown in Figure 3. In the analysis, the symmetrical plane of two HOPG sheets is selected as zero-plane, and the upward direction is set as positive. When *L*_2_ is less than 6.2 mm, the resultant force curve has only one point. The numbers of zero point are increased to three when the gap *L*_2_ is in the range of 6.2–7.0 mm. There are two zero points when *L*_2_ is equal to 7.0 mm. Zero resultant force indicates that the floating magnet can achieve an equilibrium state at these positions, but it does not mean that the floating magnet can realize a stable levitation. The levitation characteristic of the floating magnet cannot be exactly determined by the resultant force, which needs to refer to the potential energy of the floating magnet.

Figure 4 shows the potential energy of the floating magnet in the cases shown in Figure 3. According to the principle of minimum potential energy, a system will be in a stable equilibrium state when its potential energy reaches a local minimum. For the diamagnetically stabilized levitation structure, the local minimum of the potential energy does not always occur at these positions where the resultant force is equal to zero. Therefore, the floating magnet cannot be stably levitated at all the positions with zero resultant force. When *L*_2_ is less than 6.2 mm, the potential energy curve has only one local minimum, which means the floating magnet can only be stably levitated at one position. Moreover, the stable levitation position is in the zero-plane, and this state is named symmetric monostable levitation. When *L*_2_ is equal to 6.6 mm, two different minimum points appear on the potential energy curve, which indicates the floating magnet has two different stable levitation positions. In addition, two minimum points are not in the zero-plane. In the zero-plane, the floating magnet can also reach an equilibrium state because of the zero resultant force. However, this equilibrium state is easily broken by a slight external disturbance, which leads to a non-stable equilibrium. The feature with two stable levitation points is also known as bistable levitation. Adjusting *L*_2_ to 7.0 mm, the stable levitation point above zero-plane will disappear due to the large gradient of the magnetic field near the lifting magnet. In this case, the floating magnet can only be levitated below zero-plane. The phenomenon is termed asymmetric monostable levitation.

To verify the simulation results, an experimental setup was put up, which is shown in Figure 5. Two support sheets are mounted on two precision adjustment tables installed on an aluminum plate, which is used to support the lifting magnet and the upper HOPG sheet. The lower HOPG sheet is directly fixed on the aluminum plate, the upper HOPG sheet is mounted on the lower face of the support sheet, and the lifting magnet is located on the upper face of the other support sheet. *L*_1_ and *L*_2_ can be adjusted by two precision adjustment tables. *L*_1_ is the distance between the lifting magnet and the floating magnet.

Three different levitation states were verified one by one by adjusting the gap *L*_2_ between two HOPG sheets, which are shown in Figure 6. In the symmetric monostable levitation state, the floating magnet can always return to the initial position under an impact excitation. When an impact excitation is applied to the system with a bistable levitation state, the floating magnet may jump between the two equilibrium points and eventually stop at one point. In the asymmetric monostable levitation state, the floating magnet will vibrate near the equilibrium point when a slight impact excitation is adopted. Increasing the intensity of the external excitation, the floating magnet will pass through the zero-plane and be firmly adsorbed on the upper HOPG sheet due to the magnetic traction Fm. Among the three levitation states, the stability of the symmetric monostable levitation state is the best. Hence, the structure of the symmetric monostable levitation state is more suitable for developing new applications, such as sensors, actuators, and vibration energy harvesters.

## 4. Analysis of Maximum Moving Space σ

In the symmetric monostable structure, the maximum of the gap *L*_2_ is a key parameter. The maximum moving space σ derived from *L*_2_ is an important indicator of structural performance. It represents the movable space of the floating magnet in the vertical direction. It is numerically equal to the subtraction between *L*_2_ and the thickness of the floating magnet. The maximum moving space σ is determined by the part parameters, which include coating thickness, residual flux density, and structure size. Since the coating thickness and residual flux density are provided by the manufacturer, we just focus on the effect of the structure parameters on σ. To analyze the influence of structure parameters, the Taguchi method [19,20,21,22,23] is an excellent tool and is adopted in this analysis. The orthogonal array is used in the Taguchi method to arrange an experiment, which is composed of factors and levels. The experiment results are classified into three different categories by signal-to-noise (S/N) ratio: the larger-the-better (LB), the nominal-the-better (NB), and the smaller-the-better (SB). In the S/N ratio, the signal represents the desired value, whereas noise represents the undesired measured value. The S/N ratio is a parameter that can be used to evaluate the sensitivity of a parameter on the physical behavior, which is widely used to indicate the engineering quality. A larger S/N ratio corresponds to a better quality of a system. In this analysis, the objective is to maximize moving space σ of the symmetric monostable system; the LB criterion is adopted. The S/N ratio in terms of the maximum moving space σ is expressed as:(10)$$\frac{S}{N}=-10\log_{10}\left(\frac{1}{y^{2}}\right)$$
where *y* is the maximum value of moving space σ.

Seven control factors, consisting of the thickness, inner diameter, and outer diameter of the lifting magnet; the thickness, inner diameter, and outer diameter of the floating magnet; and the thickness of the HOPG sheets, along with three levels, are taken into account. These structural parameters and corresponding levels are listed in Table 2. For a Taguchi approach with 7 factors and 3 levels, a typical orthogonal array L_27_ (3^7^) with 27 runs is given in Table 3. The maximum value of the moving space σ and corresponding S/N ratios are listed in Table 3.

### 4.1. Analysis of Variance

The analysis of variance (ANOVA) is performed for moving space σ, as listed in Table 4, to evaluate the contribution of the factors. The P-magnitude of the control factors declares the statistical significance to the confidence level of 0.95 [19]. The *P* value infers that the relevant parameters of the lifting magnet have an insignificant effect on the maximum moving space of the floating magnet (*P* > 0.05). The thickness of the HOPG sheet and the structure parameters of the floating magnet are significantly related to the maximum moving space of the floating magnet (*P* < 0.01). In other words, changing the size of the lifting magnet does not cause a dramatic change in the maximum moving space σ when the floating magnet and HOPG remain unchanged. The F value of the thickness of diamagnetic material, possessing a value of 22.95, establishes the thickness of HOPG as the most significant factor. From the analysis results, the influence factors of the maximum moving space are similar to those of the maximum gap analyzed by Simon M D et al. [18].

### 4.2. Optimal Parameter Settings

Table 5 shows the mean value of S/N at each level corresponding to each factor. The effect value is the difference between the maximum and minimum S/N values of the factor at different levels. The importance of each factor affecting the maximum moving space σ can be evaluated by the effect value, and the corresponding rank is also listed in Table 5. In addition, the effect of each factor on the S/N ratio is also illustrated in Figure 7. The optimal combination of parameters is A1B3C1D1E1F3G3.

Besides, to verify the accuracy of the model, the 28th set of experiments was set according to the optimal structural parameters. MINITAB software predicted the maximum value of the moving space in the symmetric monostable structure to be 2.91 mm. The same optimization parameters are selected for the structural parameters to simulate and solve, and the maximum value of moving space σ obtained is 2.78 mm. The error between the predicted value and the simulation value is only 4.467%, which is within the acceptable range.

## 5. Energy Harvesting Experiment

The diamagnetically stabilized levitation structure is adopted as the key component of an electromagnetic energy harvester with two coils fixed on the two pyrolytic graphite sheet. The whole device is packed within a shell. External excitation is applied to the shell of the harvester to make the internal floating magnet vibrate in the horizontal direction, and induced voltage is generated within the coils to realize the vibration energy harvesting. Based on parameters of the maximum space in the abovementioned symmetric monostable structure, the structure was decided in the experiments. Since magnetic flux gradient in the horizontal direction is significantly reduced when the floating magnet has a relatively large aperture, the inner diameter parameter of the levitation magnet is selected to be zero for energy harvesting experiments. The parameters of the experiment prototype are finally determined to be A1B1C1D2E1F3G3. The experiment setup is shown in Figure 8. For the specified dimension, the model predicts the maximum moving space to be 2.91 mm, and the measured one is 2.98 mm, with only 2.34% error. Through a vibration exciter (LT-50-ST250; ECON) connected with an acceleration sensor (EA-YD-188; ECON), vibration excitation is applied to the shell of the energy harvester. An oscilloscope (MOD3014) is used to measure the voltage signal generated at both ends of the coil.

According to the maximum moving space under the structural parameters, the selected coil parameters are 0.06 mm wire diameter, 5 mm inner coil diameter, 24.5 mm outer diameter, coil thickness about 0.72 mm, and the measured coil resistance is 724 Ω. The coil is only arranged on the lower part of the upper HOPG. To indicate the moving space of the floating magnet after the coil is arranged on both sides, the paper with the same thickness as the upper coil is arranged on the lower side, as shown in Figure 9. The graphite plates with a low friction coefficient are used as the moving guide to ensure horizontal excitation without any additional load is exerted on the exciter. When the excitation peak value is set as 8 mm, the open-circuit voltage RMS at different frequencies is shown in Figure 10. When the excitation frequency is 2.6 Hz, the maximum voltage RMS reaches 250.69 mV and the power is 86.8 μW. The voltage waveform at the frequency is shown in Figure 11. If the coils are arranged on both sides of the floating magnet, the output voltage and power will be doubled. For the energy harvester, if the coil is directly connected to the Cockcroft–Walton cascade voltage doubler circuit [24] in the acquisition frequency range, the induced AC is rectified and boosted. In the experiment, an LED could be lit on after working for about 10 s, as shown in Figure 9. This excitation can be obtained by hand shaking.

## 6. Conclusions

In this paper, the levitation characteristic of a diamagnetically stabilized levitation structure was investigated theoretically and experimentally. Three different stable levitation states were found by adjusting the gap between the two HOPG sheets, which includes the symmetric monostable levitation state, bistable levitation state, and asymmetric levitation state. The influence of structure parameters on the maximum moving space between two HOPG sheets in the symmetric monostable levitation structure was studied by the Taguchi method. According to the analysis, the maximum value of the moving space is mainly affected by the floating magnet and HOPG sheet. The thickness of HOPG sheets is the most important influence factor. Besides, the optimal combination of structure parameters is also determined. Through the prediction and verification of the optimized results, the accuracy of the model is proved. Using this analysis method, an optimal diamagnetically stabilized structure was built for actuating and sensing applications. A vibration energy harvester prototype was built based on the selected parameters. Experiments were carried out to verify the low-frequency performance of the energy harvester with maximum RMS voltage 250.69 mV and 86.8 μW power under 2.6 Hz excitation.

Compared with a model previously constructed by Ding et al. [13], the output power of the energy harvester was found to be increased by about 273.3%. The analysis and experimental results show that this method is effective for guiding the structural parameters. In future research work, the dynamic model of the energy harvester can be improved, coupled with the static model, and the output characteristics of the energy harvester can be optimized more directly.

## Figures and Tables

**Figure 1 micromachines-12-00982-f001:**
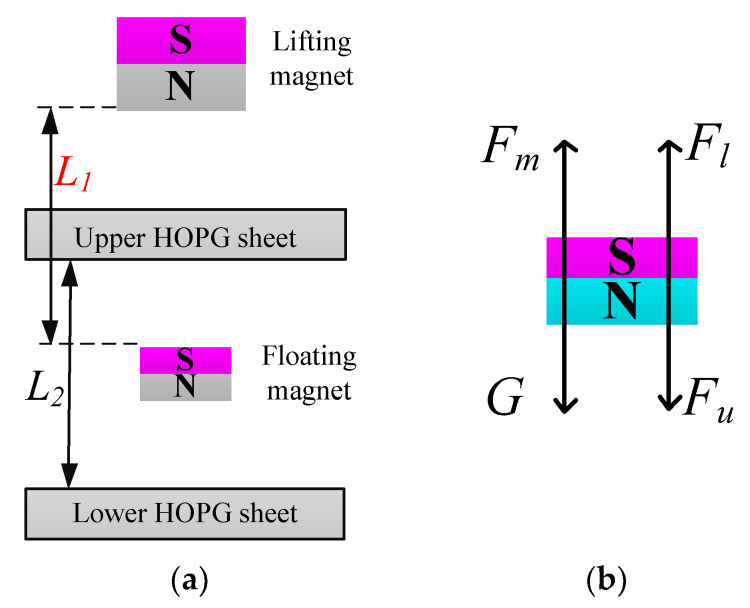
(**a**) Schematic of the diamagnetically stabilized levitation; (**b**) Mechanic analysis of the floating magnet.

**Figure 2 micromachines-12-00982-f002:**
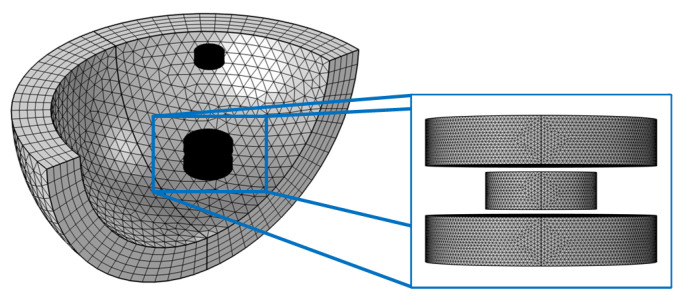
FEA model of the diamagnetically stabilized levitation.

**Figure 3 micromachines-12-00982-f003:**
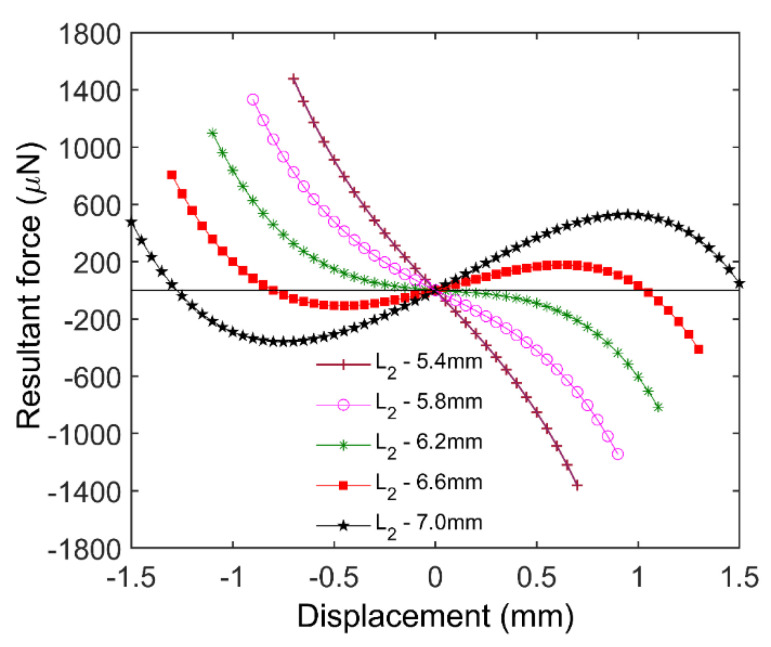
Resultant force with respect to various displacements for different gaps *L*_2_.

**Figure 4 micromachines-12-00982-f004:**
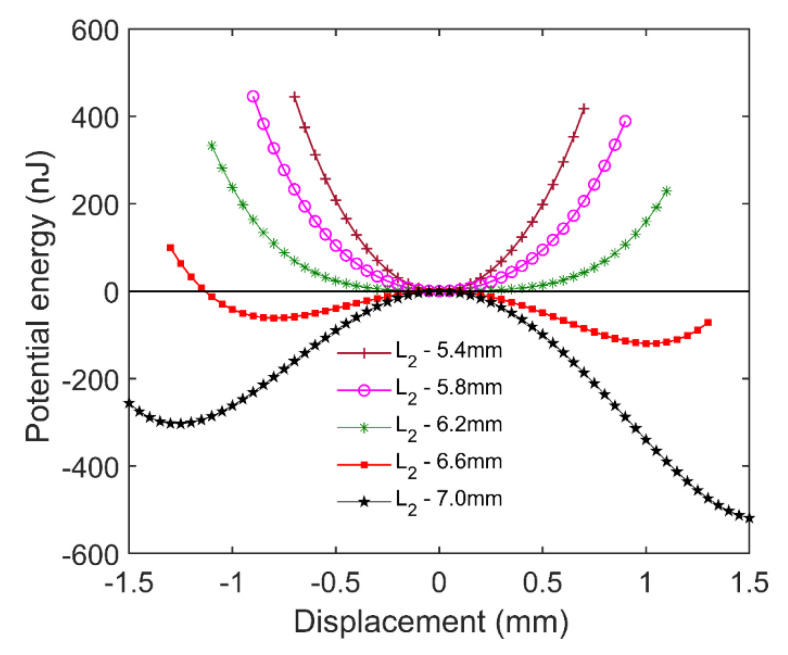
Potential energy with respect to various displacements for different gaps *L*_2_.

**Figure 5 micromachines-12-00982-f005:**
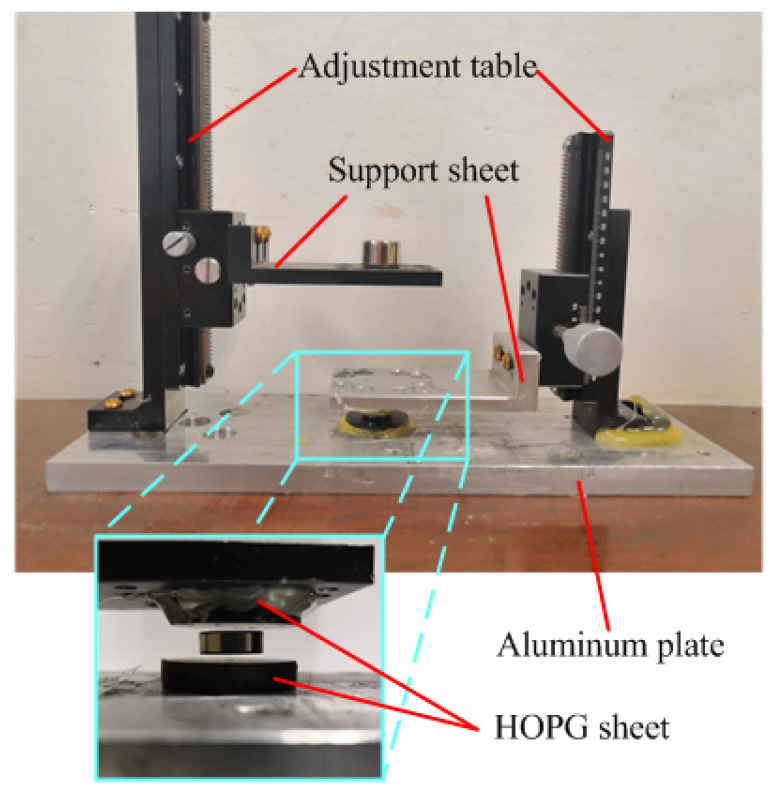
Experiment setup.

**Figure 6 micromachines-12-00982-f006:**
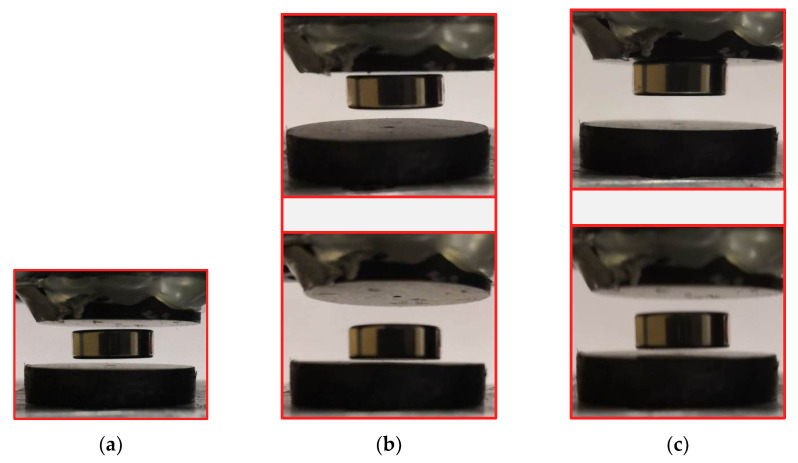
(**a**) Symmetric monostable levitation state; (**b**) bistable levitation state; (**c**) asymmetric monostable levitation state.

**Figure 7 micromachines-12-00982-f007:**
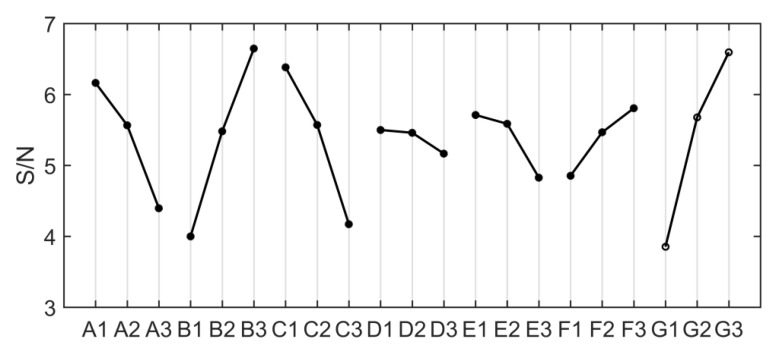
The larger the better S/N graph for the maximum moving space σ.

**Figure 8 micromachines-12-00982-f008:**
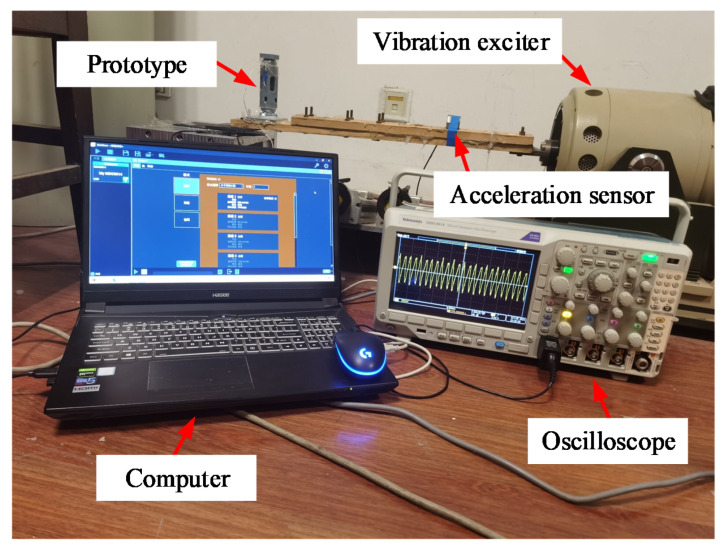
Experimental setup.

**Figure 9 micromachines-12-00982-f009:**
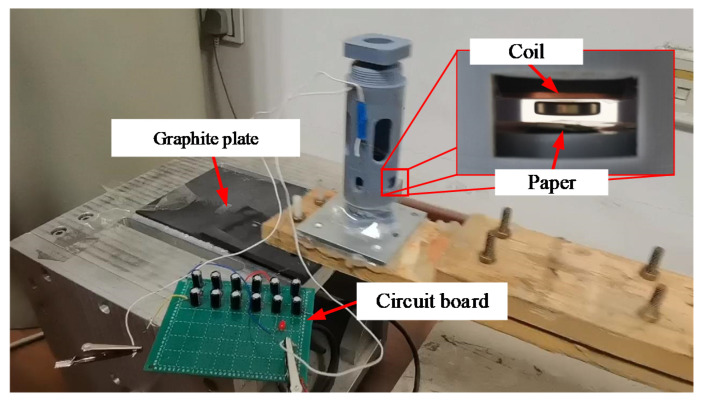
An LED illuminated by the energy harvester prototype.

**Figure 10 micromachines-12-00982-f010:**
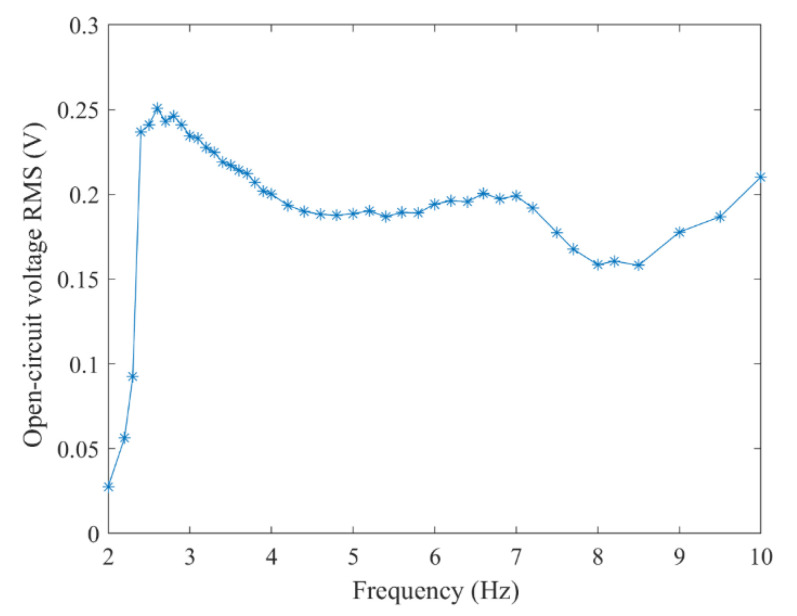
The RMS of voltage under different frequency excitation.

**Figure 11 micromachines-12-00982-f011:**
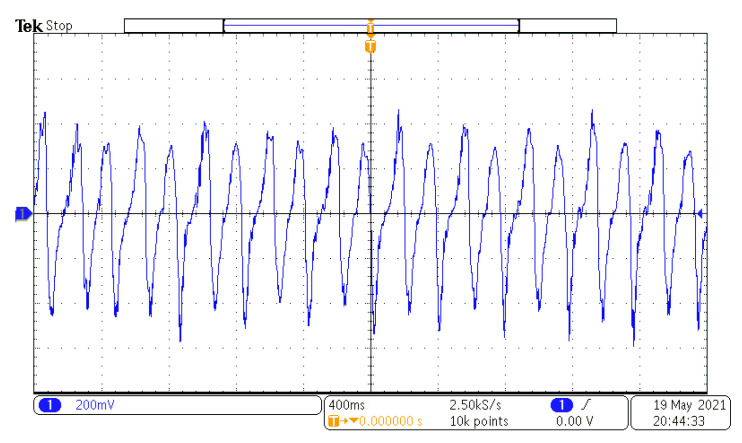
Voltage waveform at 2.6 Hz.

**Table 1 micromachines-12-00982-t001:** Structure parameters of the diamagnetically stabilized levitation.

Parameter	Value/Material
Lifting magnet and floating magnet	NdFeB-52
Residual flux density of floating magnet(T)	1.45
Residual flux density of lifting magnet(T)	1.45
Size of lifting magnet(mm)	Φ 15 × 6.35
Size of floating magnet(mm)	Φ 12 × 4
Size of diamagnetic sheets(mm)	Φ 25 × 5
Material of diamagnetic sheets	HOPG
Density of floating magnet (kg/m^3^)	7.5 × 10^3^
Magnetic susceptibility *χ* of HOPG	[8, 8, 45] × 10^−5^

**Table 2 micromachines-12-00982-t002:** Structure parameters and their levels.

Symbol	Factor (mm)	Level 1	Level 2	Level 3
A	Outer diameter of floating magnet	10	12	15
B	Inner diameter of floating magnet	0	3.175	6.35
C	Thickness of floating magnet	2	4	5
D	Outer diameter of lifting magnet	10	12.7	15
E	Inner diameter of lifting magnet	0	3.175	6.35
F	Thickness of lifting magnet	3.175	6.35	10
G	Thickness of HOPG	1	3	5

**Table 3 micromachines-12-00982-t003:** Experimental layout using an L27 orthogonal array.

Number	A	B	C	D	E	F	G	σ (mm)	S/N
1	1	1	1	1	1	1	1	1.54	3.7504
2	1	1	1	1	2	2	2	2.18	6.7691
3	1	1	1	1	3	3	3	2.30	7.2345
4	1	2	2	2	1	1	1	1.70	4.6089
5	1	2	2	2	2	2	2	2.26	7.0821
6	1	2	2	2	3	3	3	2.50	7.9588
7	1	3	3	3	1	1	1	1.78	5.0084
8	1	3	3	3	2	2	2	2.10	6.4443
9	1	3	3	3	3	3	3	2.14	6.6082
10	2	1	2	3	1	2	3	2.26	7.08217
11	2	1	2	3	2	3	1	1.44	3.16725
12	2	1	2	3	3	1	2	1.30	2.27887
13	2	2	3	1	1	2	3	2.04	6.1926
14	2	2	3	1	2	3	1	1.46	3.2870
15	2	2	3	1	3	1	2	1.64	4.2968
16	2	3	1	2	1	2	3	2.62	8.3660
17	2	3	1	2	2	3	1	2.40	7.6042
18	2	3	1	2	3	1	2	2.46	7.8187
19	3	1	3	2	1	3	2	1.50	3.5218
20	3	1	3	2	2	1	3	1.46	3.2870
21	3	1	3	2	3	2	1	0.88	−1.1103
22	3	2	1	3	1	3	2	2.00	6.0206
23	3	2	1	3	2	1	3	2.00	6.0206
24	3	2	1	3	3	2	1	1.56	3.8624
25	3	3	2	1	1	3	2	2.20	6.8484
26	3	3	2	1	2	1	3	2.14	6.6082
27	3	3	2	1	3	2	1	1.68	4.5061

**Table 4 micromachines-12-00982-t004:** Analysis of Variance for SN ratios for maximum moving space.

Source	DF	Seq SS	Adj SS	Adj MS	F	*P*
Outer diameter of the floating magnet	2	14.537	14.5375	7.2687	9.52	0.003
Inner diameter of the floating magnet	2	31.706	31.7057	15.8529	20.76	0.000
Thickness of the floating magnet	2	22.544	22.5441	11.272	14.76	0.001
Outer diameter of the lifting magnet	2	0.597	0.5971	0.2986	0.39	0.685
Inner diameter of the lifting magnet	2	4.106	4.1057	2.0528	2.69	0.108
Thickness of the lifting magnet	2	4.195	4.1951	2.0976	2.75	0.104
Thickness of HOPG	2	35.042	35.0425	17.5212	22.95	0.000
Residual Error	12	9.163	9.1632	0.7636		
Total	26	121.891				

**Table 5 micromachines-12-00982-t005:** S/N value of each factor and level.

Level	A	B	C	D	E	F	G
1	6.163	3.998	6.383	5.499	5.711	4.853	3.854
2	5.566	5.481	5.571	5.460	5.586	5.466	5.676
3	4.396	6.646	4.171	5.166	4.828	5.806	6.595
Effect	1.767	2.648	2.212	0.333	0.883	0.953	2.742
Rank	4	2	3	7	6	5	1

## Data Availability

Not applicable.

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
