# Peer review of "Levitation Characteristics Analysis of a Diamagnetically Stabilized Levitation Structure"

_micromachines, 2021, doi:10.3390/mi12080982_

Round 1
Reviewer 1 Report
The corresponding author has many earlier papers on the subject. They need to include it in the introduction and discuss what additional work is being addressed in this paper in relation to their previous work.
For instance, briefly explain how this work is different from their earlier paper: Ding, J., Gao, J., Su, Y., Zhang, K., & Duan, Z. (2018). A diamagnetically levitated vibration energy harvester for scavaging the horizontal vibration. Materials Research Express, 6(2), 025506.
Some other major concerns:
1) It is unclear to me why the optimization study was performed to increase the levitation gap when the final energy harvester was excited in the perpendicular (horizontal) direction. How does the levitation gap in the vertical direction have an influence to the vibration of the magnet in the horizontal direction?
2) It is important to emphasize that the optimal parameters mentioned in the paper are a for a specific set of lifting magnet, floating magnet and pyrolytic graphite plate. True optimal parameters are all relative ratio's, as shown in Ref [9].
Reviewer 2 Report
The authors should polish the paper suitably. The whole paper should be reviewed carefully, in order to correct all the typing errors.
In introduction, it is not enough to state the current work. It should be expended and reconstructed. Including the motivation, the main difficulties, the main work and the improvements compared with previous related works should be emphasized in this section.
The novelty of the proposed method should be highlighted carefully.
Some new advances in this field are missing, such as:
https://doi.org/10.1016/j.jfranklin.2018.03.013
https://doi.org/10.3390/app11125330
The importance of the problem considered in this paper should be further addressed.
The directions to further and improve the work should be added as future recommendation section after ‘conclusions’ section.
Round 2
Reviewer 1 Report
All comments have been addressed.